# Anomalous Pressure Effects on the Electrical Conductivity of the Spin Crossover Complex [Fe(pyrazine){Au(CN)₂}₂]

**Andrei-Cristian Gheorghe** [1,†], **Yurii S. Bibik** [2,3,†], **Olesia I. Kucheriv** [2,3], **Diana D. Barakhtii** [2,3], **Marin-Vlad Boicu** [1], **Ionela Rusu** [1], **Andrei Diaconu** [1], **Il'ya A. Gural'skiy** [2,3], **Gábor Molnár** [4,*] and **Aurelian Rotaru** [1,*]

[1]  Faculty of Electrical Engineering and Computer Science & MANSiD Research Center, Stefan cel Mare University, 13, Str. Universitatii, 720229 Suceava, Romania; andrei.gheorghe@student.usv.ro (A.-C.G.); marin.boicu@student.usv.ro (M.-V.B.); ionelar@eed.usv.ro (I.R.); andrediaconu@gmail.com (A.D.)

[2]  Department of Chemistry, Taras Shevchenko National University of Kyiv, 64/13 Volodymyrska St., 01601 Kyiv, Ukraine; yurii.bibik@univ.kiev.ua (Y.S.B.); olesia.kucheriv@univ.kiev.ua (O.I.K.); diana.barakhtii@univ.kiev.ua (D.D.B.); illia.guralskyi@univ.kiev.ua (I.A.G.)

[3]  UkrOrgSyntez Ltd., 67 Chervonotkatska St., 02094 Kyiv, Ukraine

[4]  LCC, CNRS and Université de Toulouse, UPS, INP, F-31077 Toulouse, France

*  Correspondence: gabor.molnar@lcc-toulouse.fr (G.M.); rotaru@eed.usv.ro (A.R.)

†  These authors have equally contributed.

**Abstract:** We studied the spin-state dependence of the electrical conductivity of two nanocrystalline powder samples of the spin crossover complex [Fe(pyrazine){Au(CN)₂}₂]. By applying an external pressure (up to 3 kbar), we were able to tune the charge transport properties of the material from a more conductive low spin state to a crossover point toward a more conductive high spin state. We rationalize these results by taking into account the spin-state dependence of the activation parameters of the conductivity.

**Keywords:** spin crossover; charge transport; high pressure

## 1. Introduction

The spin crossover (SCO) phenomenon is observed in a large number of transition metal complexes with a $3d^4$–$3d^7$ electron configuration [1]. However, Fe(II) is by far the most commonly studied metal ion. For these complexes, depending on the ligand field strength, the central Fe(II) ion may exist in two different electronic configurations: low spin (LS, S = 0) at lower temperatures and high spin (HS, S = 2) at higher temperatures. The conversion from one state to the other can be induced by different external stimuli, such as temperature, pressure, and light irradiation.

In the last decade, an important number of papers focused on the charge transport properties of spin crossover (SCO) materials at different scales: bulk [2–7], single molecule [8–12], nano- and micro-particles [13–16], composites [17], and thin films [18–21], in relation to interesting perspectives for the application of these compounds in molecular electronics and spintronics. Many of these findings are covered in recent reviews [22–25].

In particular, many interesting results have been reported about the charge transport properties of the benchmark [Fe(Htrz)₂(trz)](BF₄) (trz = triazolato) SCO complex. However, opposite conductivity changes with respect to the spin state of the complex have been reported. In particular, measurements on nanoelectronic devices based on a single SCO nanoparticle showed a more conducting HS state [13]. The reverse situation has been observed in general for particle assemblies [2–6,14–16]. This intriguing

difference arises most probably due to the different charge transport mechanisms involved in the different experiments (e.g., tunneling vs. hopping). However, it was predicted that, even in the frame of the same transport mechanism, the sign of the conductivity switching may be inversed. In particular, by extrapolating the temperature dependence of the electrical conductivity $\sigma$ of $[Fe(Htrz)_2(trz)](BF_4)$ in both spin states, we have shown that the two $\sigma(T)$ curves are crossing at high temperatures, and we suggested that this crossing point might be reached under an applied external pressure [2]. Indeed, it is well known that an applied external pressure stabilizes the LS state due to its lower molecular volume, shifting the spin transition toward higher temperatures typically by 10–20 K·kbar$^{-1}$ [26]. On the other hand, the activation parameters of the conductivity are not altered much by applying moderate external pressures (~kbar) [5]. Then, in some particular circumstances, we can expect an inversion between the conductivity of the two spin states as a function of the applied pressure.

In the case of the $[Fe(Htrz)_2(trz)](BF_4)$ complex, this inversion is difficult to approach experimentally due to the rather high temperatures involved. In this paper, we demonstrate that it is, however, possible for the SCO complex $[Fe(pyrazine)\{Au(CN)_2\}_2]$. This three-dimensional (3D) coordination network displays an abrupt spin transition around 360 K, accompanied by a hysteresis loop [27]. We show that it is possible to tune the electrical conductivity of the material, going from a substantially more conductive LS state (at atmospheric pressure) to a crossover point denoting the onset of a more conductive HS state at high pressures. These results come to substantiate our previous analysis done on the $[Fe(Htrz)_2(trz)](BF_4)$ system and highlight the complex interplay between the charge transport and spin state in SCO materials.

## 2. Results

Two powder samples of $[Fe(pyrazine)\{Au(CN)_2\}_2]$ were synthesized following the synthetic procedure described in [27], except that an excess of pyrazine was used (5 equivalent for sample **S1** and 2 equivalent for sample **S2**). The influence of temperature and pressure on the charge transport and spin transition properties of the two samples was investigated by means of magnetic susceptibility measurements and broadband dielectric spectroscopy. The effect of synthetic procedure on the morphology and composition of the samples was assessed using elemental analysis, FTIR spectroscopy, and scanning electron microscopy (SEM). The SEM analysis of the two samples revealed rod shaped nanoparticles with a mean length of about 140 and 200 nm and a mean diameter of 60 and 70 nm for **S1** and **S2**, respectively (Figure 1a,b).

The chemical identity of the particles has been confirmed by their magnetic properties (Figure 1c), elemental analysis (see the Supporting Information, SI), and vibrational spectra (Figure 1d). The temperature dependence of the product of the molar magnetic susceptibility and the temperature ($\chi T$) was investigated under a DC magnetic field of 1000 Oe with a temperature sweep rate of 2 K·min$^{-1}$. Both samples exhibit spin transition with a thermal hysteresis loop of ca. 17 K width centered around 359 (**S1**) and 362 K (**S2**), respectively, but the hysteresis is more rectangular for sample **S1** (Figure 1c), indicating possibly better crystallinity and/or better sample homogeneity. The $\chi T$ value of ca. 3.6 cm$^3$mol$^{-1}$K at high temperatures corresponds to a fully populated HS state (S = 2), whereas at low temperatures, the LS state (S = 0) is also fully populated, the $\chi T$ value in this spin state being close to zero for both samples.

The pressure effect on the electrical conductivity of the two samples was analyzed using a broadband dielectric spectrometer (100 mHz–1 MHz) as a function of temperature (25–250 °C) at fixed pressure values (up to 3 kbar). Silicone oil was used as an inert, hydrostatic pressure transmitting medium. Figure 2 shows the temperature dependence of the real part of the AC electrical conductivity ($\sigma'$), the real part of the dielectric permittivity ($\varepsilon'$), and the imaginary part of the electric modulus ($M''$), recorded on sample **S1**, under various constant pressure values at an AC frequency of 100 Hz. (N.B. The frequency dependence of $\sigma'$, $\varepsilon'$, and $M''$ at various temperature and pressure values is shown for both samples in the SI.)

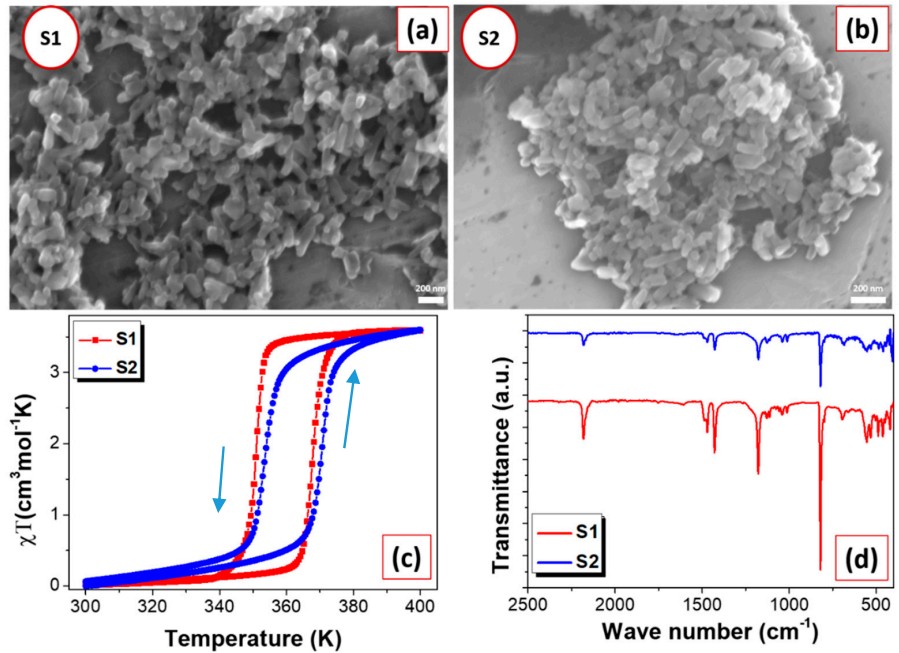

**Figure 1.** Sample characterization: (**a**,**b**) SEM micrographs, (**c**) temperature dependence of the magnetic susceptibility × temperature product on heating and cooling, and (**d**) FTIR spectra of samples **S1** and **S2**.

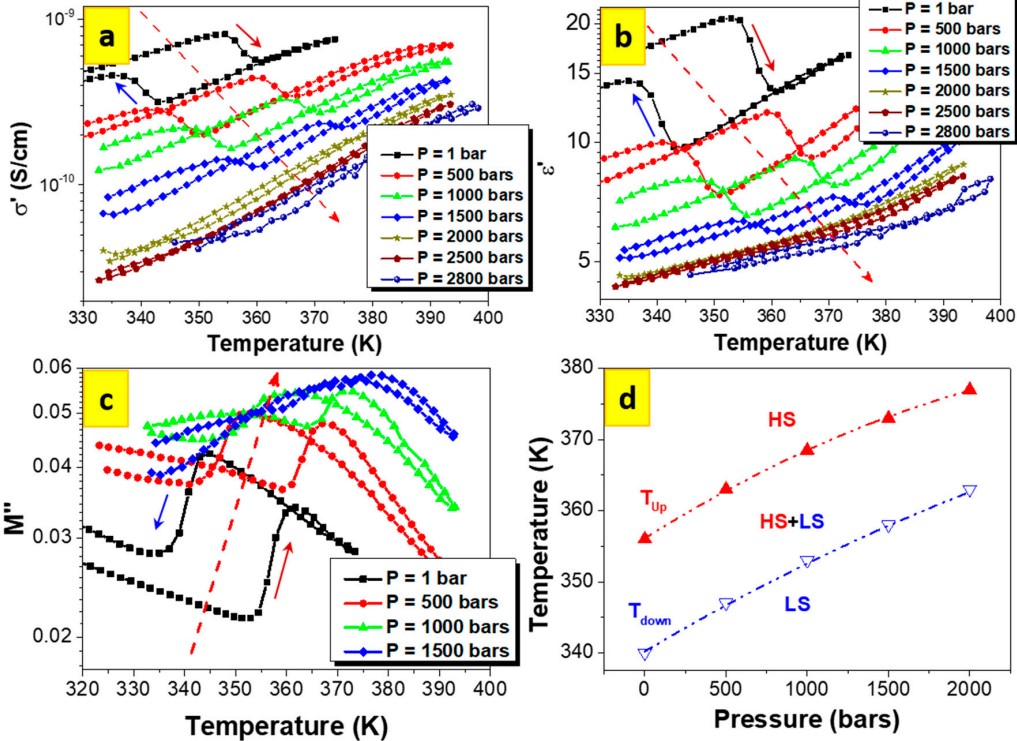

**Figure 2.** Temperature dependences of (**a**) the real part of the electrical conductivity; (**b**) the real part of the dielectric permittivity; and (**c**) the imaginary part of the electric modulus, recorded at 100 Hz applied electric field frequency for various applied pressures for sample **S1**. (**d**) *P*,*T*-phase diagram extracted from the electrical measurements. (Dashed lines are guides to the eye).

At ambient pressure, this sample exhibits a thermal hysteresis in its electrical properties associated with the spin crossover phenomenon. Similar to previous reports, the LS state is more conductive when compared to the HS state (at 350 K $\sigma_{LS} = 7.6 \times 10^{-10}$ S·cm$^{-1}$ and $\sigma_{HS} = 3.8 \times 10^{-10}$ S·cm$^{-1}$).

For increasing applied pressures, the hysteresis width remains nearly constant, but the spin transition temperatures are upshifted by ca. 13 K·kbar$^{-1}$. This Clapeyron slope falls in the range typically encountered for SCO complexes [26]. (N.B. There is a deviation from linearity at higher pressures, but in this pressure range, the transition temperatures can be assessed with high uncertainty.) These results are in good agreement with the magnetic measurements under external pressure (see the SI).

The striking observation in Figure 2 is that the amplitude of the electrical conductivity and dielectric permittivity switching between the two spin states, i.e., $(\sigma_{LS} - \sigma_{HS})/\sigma_{HS}$ and $(\varepsilon_{LS} - \varepsilon_{HS})/\varepsilon_{HS}$, drastically decreases with increasing pressure and vanishes around ca. 2000 bar. As a result, for high pressures, the hysteresis disappears, and the material is characterized by the same values of electrical conductivity and dielectric permittivity in the two spin states. This unexpected observation for [Fe(pyrazine){Au(CN)$_2$}$_2$] is in stark contrast to what was reported for [Fe(Htrz)$_2$(trz)](BF$_4$), in which case actually a considerable increase of both the conductivity and the switching amplitude was reported under an applied pressure [5]. The vanishing of the conductivity switching amplitude for [Fe(pyrazine){Au(CN)$_2$}$_2$] is progressive, which refutes the hypothesis of a pressure induced structural transition at the origin of this phenomenon. It is also not an irreversible modification of the compound because upon decompression, the initial properties were restored. (It is important to note also that the results reported in Figure 2 have been reproduced several times with the same sample and also with new portions from the same synthesis batch.) Taking into account all these ingredients, we suggest that the smearing out of the conductivity switching under pressure in this sample is linked to the interplay between the conductivity activation parameters and the pressure tuning of the spin transition temperatures.

The activation parameters of the electrical conductivity in the two spin states were determined using the Arrhenius relationship:

$$\sigma = \sigma_0 e^{-\frac{E_a}{k_B T}} \tag{1}$$

where $\sigma_0$ is the pre-exponential factor, $E_a$ is the thermal activation energy, and $k_B$ is the Boltzmann constant. As shown in Table 1, the activation energy is lower in the LS state for the whole pressure range. An important role in the charge transport properties is played by the pre-exponential factor $\sigma_0$, which depends on the competition of two terms: the hopping distance (which is larger in the HS state) and hopping frequency (which is higher in the LS state). Indeed, in the case of hopping transport, the Einstein diffusion relation is often used to connect the DC conductivity with the hopping frequency:

$$\sigma_{dc} = \frac{n_c (ea)^2}{6 k_B T} \vartheta_p = \frac{n_c (ea)^2}{6 k_B T} \vartheta_{0p} exp\left[\frac{-E_p}{k_B T}\right] \tag{2}$$

where $n_c$ is the carrier density, $e$ is the electronic charge, $a$ is the hopping distance, $v_p$ is the hopping frequency, $v_{0p}$ is the relevant phonon frequency, and $E_p$ is the activation energy of the hopping process. In general, one can assume that $E_p = E_a$ [28].

**Table 1.** Activation parameters of the AC conductivity (1 Hz) in the low spin (LS) and high spin (HS) states for sample **S1**.

| Pressure (bar) * | $\sigma_0^{LS}$ (S/m) | $\sigma_0^{HS}$ (S/m) | $E_a^{LS}$ (eV) | $E_a^{HS}$ (eV) |
|---|---|---|---|---|
| 1 | 1.1 (9) × 10$^{-4}$ | 6.3 (3) × 10$^{-3}$ | 0.22 (2) | 0.36 (2) |
| 500 | 1.3 (4) × 10$^{-3}$ | 1.1 (1) × 10$^{-2}$ | 0.31 (9) | 0.40 (1) |
| 1000 | 1.7 (2) × 10$^{-3}$ | 2.1 (1) × 10$^{-2}$ | 0.34 (1) | 0.43 (4) |
| 1500 | 8.2 (1) × 10$^{-3}$ | 1.9 (4) × 10$^{-1}$ | 0.40 (8) | 0.51 (8) |

* For higher pressures, the electrical properties of the two spin states cannot be distinguished.

Using these parameters, we extrapolated the temperature dependence of the electrical conductivity in the two spin states (Figure 3). It appears that the two $\sigma = f(T)$ curves, characterizing the temperature dependence of the electrical conductivity in the HS and LS states, cross each at a conductivity crossover

temperature ($T_{CC}$), which tends to decrease with increasing pressure. At the same time, as we have seen above, the spin transition temperature ($T_{SCO}$) monotonously increases with increasing pressure. Our experiments show that for pressures between 2 and 3 kbar, $Tcc \cong T_{SCO}$, which explains the smearing out of the conductivity switching upon the SCO.

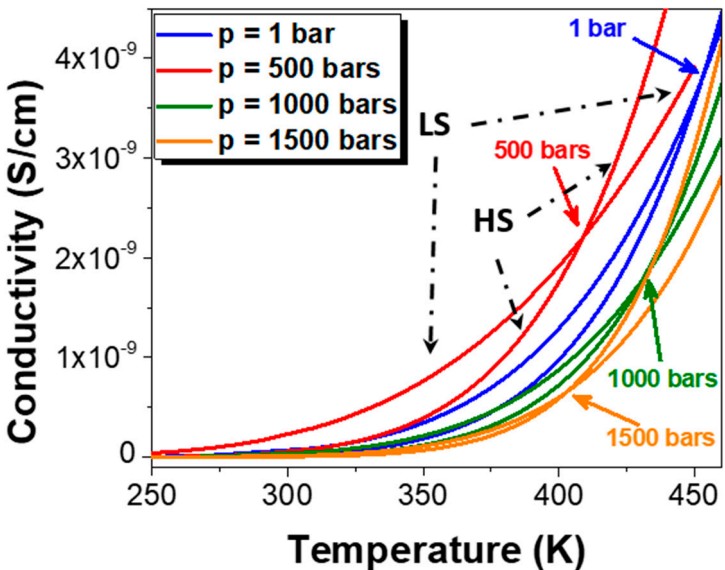

**Figure 3.** Temperature dependence of the electrical conductivity of sample **S1** in the HS and LS states for different applied pressures, simulated using the activation parameters in Table 1.

Unfortunately, our pressure setup is limited to 3 kbar, and therefore, the crossover from a more conducting LS to a more conducting HS state could not observed for sample **S1**. However, this effect becomes perceptible in sample **S2**. This sample has the same composition and similar SCO properties to sample **S1**, but it is characterized by a different microstructure and presumably different crystallinity. The temperature dependence of the electrical conductivity and dielectric permittivity of **S2** does not show any spin-state dependence at atmospheric pressure, indicating that in this sample, $Tcc \cong T_{SCO}$ is already at 1 atm pressure (see the SI). On the other hand, for applied pressures above ca. 1000 bar, a thermal hysteresis loop appears in the temperature dependence of the electrical conductivity, showing a more conducting HS state (Figure 4). The conductivity change between the two spin states is rather small in this sample (~$10^{-12}$ S/cm), which is comparable in magnitude with the conductivity drifts induced by the thermal/pressure cycling. Nevertheless, the fact that we observed systematically for each pressure value a higher conductivity in the cooling branch of the SCO hysteresis supports our hypothesis that in the case of sample **S2**, the application of pressure leads to $T_{CC} < T_{SCO}$, i.e., the HS state becomes more conducting.

As has been previously mentioned, samples **S1** and **S2** were obtained using different amounts of pyrazine ligand during the synthesis. This does not substantially affect the size of the nanocrystals and has no considerable effect on the sample composition and magnetic properties. However, there could be small differences in the crystallinity and in the number of defects (e.g., $Au(CN)_2$ vacancies), which might influence their electrical conductivity. This hypothesis is supported by the magnetic data, which show that **S1** exhibits a more abrupt spin transition than **S2**. It is worth noting also that the structure of $[Fe(pz)\{Au(CN)_2\}_2]$ is built of two identical interpenetrated 3D frameworks [27]. This does not leave any solvent available voids in the grid and makes this complex insensible to the guest effects, which are usually observed in 3D Hofmann clathrate analogues [29]. On the other hand, an important feature of $[Fe(pz)\{Au(CN)_2\}_2]$ is a huge anisotropic deformation of the wine-rack type, which takes place upon the spin transition. These deformations considerably change the geometry of the framework, which consequently may also influence the conductivity of the complex.

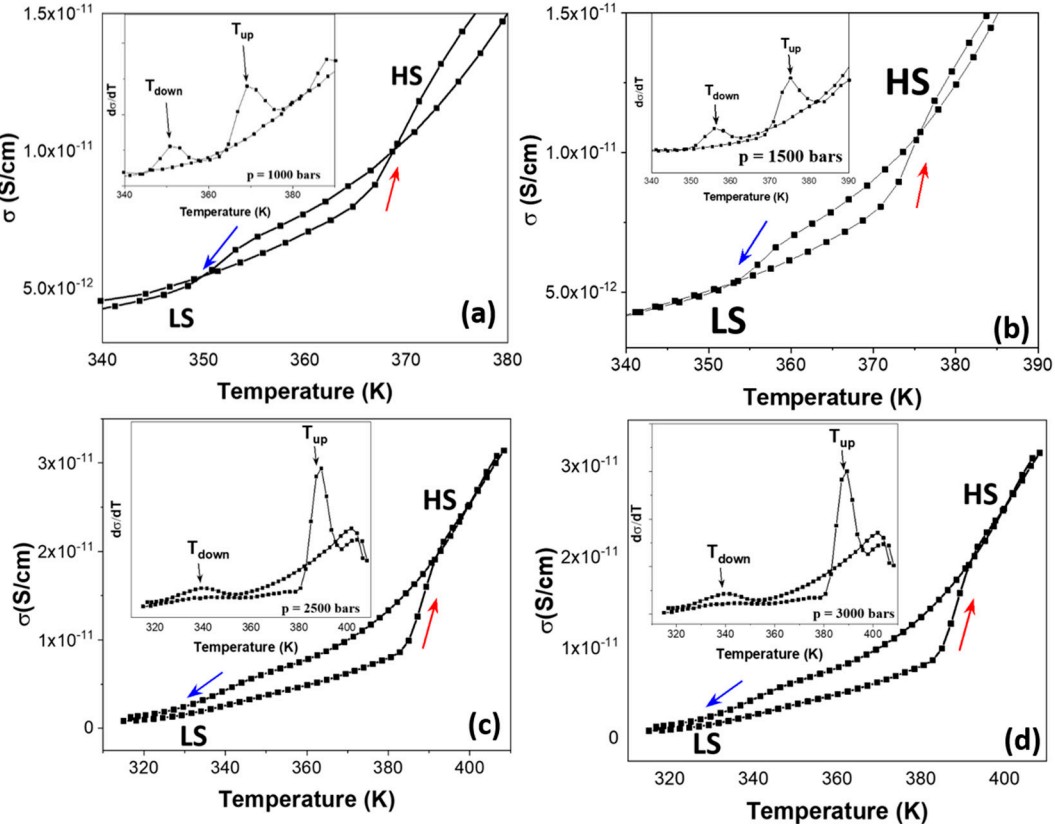

**Figure 4.** Temperature dependence of the AC electrical conductivity recorded on sample **S2** under various external pressures: (**a**) 1000 bar; (**b**) 1500 bar; (**c**) 2500 bar; and (**d**) 3000 bar. The insets show the first derivative *dσ/dT* vs. *T*.

## 3. Materials and Methods

Magnetic measurements were performed using a MPMS3 SQUID magnetometer (Quantum Design Inc., San Diego, CA, USA) in DC mode, under a DC magnetic field of 1000 Oe, with a temperature rate of 2 K/min. Data were corrected for the sample holder and sample diamagnetic contributions. Electrical conductivity measurements were performed with a CONCEPT 40 Broadband Dielectric Spectrometer (Novocontrol Technologies GmbH, Montabaur, Germany) as a function of temperature under constant pressure. The spectra were recorded with an Alpha-A high performance frequency analyzer (3 μHz–20 MHz) and a high pressure system option for dielectric measurements from 0 to 3 kbar in the 25–250 °C temperature range. The temperature was changed in sweeping mode at 0.5 K/min, in both heating and cooling modes, and the pressure was manually lowered (increased) to maintain a constant value (±5 bar) throughout the entire temperature sweep range, while continuously recording frequency spectra. FTIR spectra were recorded with a Spectrum Two spectrometer (PerkinElmer Inc., Waltham, MA, USA) in attenuated total reflectance (ATR) mode in ambient conditions. SEM micrographs were recorded in secondary electron mode using a SU-70 microscope (Hitachi Ltd., Tokyo, Japan). The particles were deposited on an Al mount from a previously sonicated suspension in toluene.

## 4. Conclusions

In summary, we have shown that by applying an external pressure it is possible to tune the electrical conductivity of the spin crossover complex [Fe(pyrazine){Au(CN)$_2$}$_2$] from a more conducting LS state (low pressure) toward a more conducting HS state (high pressure) through a crossover region wherein the charge transport properties of the two spin states cannot be distinguished. This phenomenon has been ascribed to the interplay between the conductivity thermal activation parameters and the

pressure induced shift of the spin transition temperature. The different electrical properties of the two studied samples have been tentatively ascribed to their different crystallinity, influenced by the excess of ligands used during the synthesis. Further work will be necessary to examine these phenomena as a function of the frequency of the applied electric field, which may also bring useful insight into the underlying physical mechanisms [6]. Overall, these results highlight that the spin-state dependence of the electrical conductivity and dielectric permittivity of SCO materials depends strongly on the experimental conditions and any generalization must be made with care.

**Supplementary Materials:** The following are available online at http://www.mdpi.com/2312-7481/6/3/31/s1, Figures S1–S9: Supplementary electrical and magnetic measurement data.

**Author Contributions:** Conceptualization: A.R., G.M. and I.A.G.; sample synthesis and experimental investigation: A.-C.G., Y.S.B., O.I.K., D.D.B., M.-V.B. and A.D.; activation energy analysis: I.R.; writing: A.R., G.M. and I.A.G.; funding acquisition: A.R., G.M. and I.A.G. All authors have read and agreed to the published version of the manuscript.

**Funding:** This work was funded by the European Commission through the SPINSWITCH project (H2020-MSCA-RISE-2016, Grant Agreement No. 734322). Financial support of the EXCALIBUR project (Contract No. 18 PFE/16.10.2018) and Ministry of Education and Science of Ukraine (grant No. 19BF037-01M) is also acknowledged.

**Conflicts of Interest:** The authors declare no conflict of interest.

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
