# Peer review of "Anomalous Pressure Effects on the Electrical Conductivity of the Spin Crossover Complex [Fe(pyrazine){Au(CN)2}2]"

_magnetochemistry, doi:10.3390/magnetochemistry6030031_

Round 1

Reviewer 1 Report

Authors study the impact of spin states variation on pressure dependent electrical properties in SCO coordination polymer. From the previous studies focused on the conductivity of SCO complexes is obvious that some of SCO complexes have more conductive HS and less conductive LS state, and other compounds show the opposite behavior (i.e. ref 5, 13-16 in the manuscript). Therefore authors intended to explain this inconsistency by theoretical modeling of p and T dependent HS and LS conductivity curves, which resulted in the definition of new parameter “conductivity crossover temperature”, which seems like tell us if corresponding spin state of SCO compound will be more or less conductive as the other spin state at the given pressure. I think that experimental observations are very well supported/explained by theoretical modeling, and therefore presented story deserves to be published in Magnetochemistry journal.

I also have one question, which should be addressed before the final acceptance of the paper:

Both samples S1 and S2 show very similar thermal SCO, but very different conductivity properties at various p and T, which is attributed to different microstructures (S2 has larger nanocrystals comparing to S1). Thus, my question is what is the impact of crystallinity on the conductivity and Tcc? I understand that this might be very difficult question, but at least tentative attempt to answer on it should be included into the conclusion.

Minor points:

Line 78: it is temperature dependence of chiT, not chi

Author Response

Referee 1

Comment: Both samples S1 and S2 show very similar thermal SCO, but very different conductivity properties at various p and T, which is attributed to different microstructures (S2 has larger nanocrystals comparing to S1). Thus, my question is what is the impact of crystallinity on the conductivity and Tcc? I understand that this might be very difficult question, but at least tentative attempt to answer on it should be included into the conclusion.

Reply: Samples S1 and S2 were obtained using different amounts of pyrazine ligand during the synthesis. The different synthesis conditions do not substantially affect the magnetic properties. However, there could be minor differences in the crystallinity (e.g. Au(CN)2 defects), which might influence the electrical conductivity of the samples. The fact that sample S1 exhibits a more rectangular shaped hysteresis may be a sign of better crystallinity, which can therefore explain its higher conductivity. (We have added comments to the revised MS along these lines on pages 3 and 8.)

Comment: Minor points: Line 78: it is temperature dependence of chiT, not chi

Reply: Corrected.

Reviewer 2 Report

The main message of the work „Anomalous Pressure Effects on the Electrical  Conductivity of the Spin Crossover Complex [Fe(pyrazine){Au(CN)2}2]” by A. Rotaru et al. is verification of interesting thesis that crossing point of conductivities of HS and LS forms can be reached by an application of external pressure and what is most imortant further pressure elevation should result in inversion in conductiovity and spin state.

Studies on temperature dependence of spin crossover under various applied pressures for sample S1 allowed to  determine correlation between pressure and activation energies Interpolation of temperature dependency of conductivity  indicated that location  of crossing between  conductivity vs temperature curves for HS and LS forms depends on applied pressure. It leads to conclusion that at higher pressures it is possible that  HS form will exhibite higher conductivity then the LS one.

Unfortunately the authors were not able to show their hypothesis experimantally for sample S1 because of apparature limitations. For this purpose authors exploited second sample S2. Authors declare identity of chemical compositions of S1 and S2 together with physical similarity of the samples  - the size and shape of microcrystals in samples S1 and S2 are very close. Although properties of S1 are different in relation to S2. Namely conductivity of S2 is significantly lesser in relation to S1 and at 1000 bar the temperature  difference between conductivities of HS and LS forms in S2 (in the temperature range of occurence of hysteresis) is very small – rather close to the one obobserved for 1 at higher pressures than 1000 bar.

Investigations of sample S2 revealed that at 1000 bar or at higher pressures the higher conductivity exhibites HS form. Concomitantly it can be expected that at lower pressures the higher conductivity should exhibites LS form in  S2. How it looks dependency between conductivity and temperature at ambient pressure for sample S2? Occurence of such dependency, that is higher conductivity of LS form occured for pressures below 1000 bar  in S2 also will confirm thesis of authors. Could authors refer to this?

Authors performed synthesis of samples S1 and S2 in different conditions. What was the reason to apply different synthesis conditions if physical forms are practically the same?

Differences in synthetic details  can be responsible for different properties of samples S1 and S2. I think that magnetic properties are not identical but very similar. On the one hand it can indicate that samples S1 and S2 are not chemically identical.  Can authors add results of elemental analysis for S1 and S2? On the other hand properties of spin crossover systems are strongly dependent on the properties of solid sample (crystal size, defects introduced mechanically or resulted from for example of speed of product precipitation, sample history).

In my opinion presented results are interesiting and deserve on publication in Magnetochemistry, however, in the revised version of the manuscript authors should refer to possible reasons for the differentiation in conductivity of S1 and S2.  

Author Response

Referee 2

Comment: Concomitantly it can be expected that at lower pressures the higher conductivity should exhibites LS form in  S2. How it looks dependency between conductivity and temperature at ambient pressure for sample S2? Occurence of such dependency, that is higher conductivity of LS form occured for pressures below 1000 bar  in S2 also will confirm thesis of authors. Could authors refer to this?

Reply: Thermal dependence of the electrical conductivity recorded under external pressure below 1000 bar has been inserted into the revised ESI (Fig. S1). No significant difference between the HS and LS states can be observed, which does not confirm nor contradict our thesis.

Comment: Authors performed synthesis of samples S1 and S2 in different conditions. What was the reason to apply different synthesis conditions if physical forms are practically the same? Differences in synthetic details can be responsible for different properties of samples S1 and S2. I think that magnetic properties are not identical but very similar. On the one hand it can indicate that samples S1 and S2 are not chemically identical.  Can authors add results of elemental analysis for S1 and S2?

Reply: The use of different amounts of pyrazine ligands in the synthesis was aimed for tuning the ligand field. However, it did not allow for appreciable changes of the spin transition temperature. As requested by the referee, we have added elemental analysis data to the revised ESI for the two samples (page 1). They are comparable within experimental uncertainty (such as the FTIR spectra) confirming chemical identity.

Comment: … in the revised version of the manuscript authors should refer to possible reasons for the differentiation in conductivity of S1 and S2.

Reply: Samples S1 and S2 were obtained using different amounts of pyrazine ligand during the synthesis. The different synthesis conditions do not substantially affect the magnetic properties. However, there could be minor differences in the crystallinity (e.g. Au(CN)2 defects), which might influence the electrical conductivity of the samples. The fact that sample S1 exhibits a more rectangular shaped hysteresis may be a sign of better crystallinity, which can therefore explain its higher conductivity. (We have added comments to the revised MS along these lines on pages 3 and 8.)

Reviewer 3 Report

I found the paper by Georghe et al. to be very interesting. The authors present the result of the conductivity studies for two nanmaterials of the same spin crossover complex, differing in the method of preparation and hence in the size of the nanoparticles. The conductivity measurements have been performed as the function of the temperature under different pressure. The authors report the decrease of the hysterisis observed in the conductivity measurment with the increasing pressure and ascribe this effect as resulting  from spin dependence of the acitivation parameters of conductivity. The results are pretty interesting yet the paper needs an improvement in the way how they are presented.

1. The way of presentation raises my main concern. Already in the Abstract the readers is confronted with a sentence "to tune the charge transport properties of the material 19 from a more conductive low spin state to a crossover point towards a more conductive high spin state"

I do not think that the abstract is the best place to make the reader to solve the puzzle offered by the idea of switching from something more conductive to something more conductive. The same sentence is repeated in the Conclusion yet I am still unable to understand the meaning of it.

-It can not be the problem of the conductivity increasing with the pressure for both spin states, because on the basis of Fig.2  it actually decrease with the increasing pressure.
-It can not be the problem of the conductivity of the distinguishable LS and HS states compared to the crossover region, because - if interpret Fig 2, at least for the black curve in this region the conductivity is lower than that of LS state but higher than that of the HS one.

I read this paper several times to understand what the authors mean and what is the reference that shows the lower conductivity but I am helpless. The authors shall state it clearly, otherwise the other readers can not grasp it, I am afraid.

Furthermore, the graphical representation of the result does not help to understand the flow of the reasoning.

-Firstly, while looking at Figure 2 one may ask the authors to indicate which branch of the curves presented there correpsonds to heating and cooling.
-Sceondly, I again have the problem to understand some point. Looking at the Figure 2d one may deduce that basing  on the electrical measurements the hysteris of ca 25 K is retained at all values of the pressure. Yet this is contradictory to the decrease of hysteris in sigma, epsilon and M'' curves shown in the same Figures. Please comment on that.
Additionaly, in my opinion an additional graphic showing the dependence of the molar fraction of the high-spin phase on temperature for different values of the pressure shall be shown.
-Thirdly, Figure 3. Please indicate where is the low-spin and where the high-spin state.
Concerning the results for the S2. It is really awkward that the graphics showing the results for S2 differ totally from that for S1. Any comparison requires jumping between two different sets of Figures, one coloured and one black and white. At least one Figure in which the conductivity data for S1 and S2 were presented together would be of a great help.

2. Apart from this comments on the scholarly side of the paper I have only one remark. The complex studied by the authors is a 3D one but not a typical one. In this very system there are two intercalated 3D networks of the same species. According to my knowledge, there are no conductivity studies on the "simple" 3D systems like that of [Fe(pyz){M(CN)4}] type, so it is not possible to compare them. Yet, this  peculiar character of the structure shall be mentioned.

Author Response

Referee 3

Comment: Already in the Abstract the readers is confronted with a sentence "to tune the charge transport properties of the material 19 from a more conductive low spin state to a crossover point towards a more conductive high spin state" I do not think that the abstract is the best place to make the reader to solve the puzzle offered by the idea of switching from something more conductive to something more conductive. The same sentence is repeated in the Conclusion yet I am still unable to understand the meaning of it.

Reply: This problem has not been indicated by Referees 1&2, we thus believe the message is fairly clear.

Comment: … while looking at Figure 2 one may ask the authors to indicate which branch of the curves presented there correpsonds to heating and cooling.

Reply: Figures 1 and 2 have been modified by indicating with arrows the heating and cooling branches.

Comment: Looking at the Figure 2d one may deduce that basing on the electrical measurements the hysteris of ca 25 K is retained at all values of the pressure. Yet this is contradictory to the decrease of hysteris in sigma, epsilon and M'' curves shown in the same Figures. Please comment on that.

Reply: There is no contradiction. The switching temperatures have been extracted from the temperature dependence of the first derivative of the ac conductivity (Fig. 2a), which are identical with the other electric/dielectric parameters. The hysteresis width is slightly decreasing from 16 K at 1 bar to 14 K at 2000 bars.

Comment: Additionaly, in my opinion an additional graphic showing the dependence of the molar fraction of the high-spin phase on temperature for different values of the pressure shall be shown.

Reply: We have performed magnetic measurements under pressure on the two samples, which are now added to the revised ESI (Figures S3 and S4).

Comment: Figure 3. Please indicate where is the low-spin and where the high-spin state.

Reply: Figure 3 has been modified accordingly.

Comment: It is really awkward that the graphics showing the results for S2 differ totally from that for S1. Any comparison requires jumping between two different sets of Figures, one coloured and one black and white. At least one Figure in which the conductivity data for S1 and S2 were presented together would be of a great help

Reply: A figure that compares the temperature dependence of the electrical conductivity of the two samples, recorded at 1500 bars, has been inserted in the revised ESI (Figure S2).

Question: The complex studied by the authors is a 3D one but not a typical one. In this very system there are two intercalated 3D networks of the same species. According to my knowledge, there are no conductivity studies on the "simple" 3D systems like that of [Fe(pyz){M(CN)4}] type, so it is not possible to compare them. Yet, this  peculiar character of the structure shall be mentioned.

Reply: Indeed, the structure of [Fe(pz){Au(CN)2}2] is built of two identical interpenetrated 3D frameworks, as established in ref. [27]. This does not leave any solvent available voids in the grid and makes this complex insensible to the guest effects, which are usually observed in 3D Hofmann clathrates [29]. In addition, an important feature of [Fe(pz){Au(CN)2}2] complex is a huge anisotropic deformation of the wine-rack type, which takes place upon spin transition. These deformations considerably change the geometry of the framework, which consequently may also influence the conductivity of the complex. We have added some notes on these points to the revised MS on page 5.

Round 2

Reviewer 2 Report

The authors answered my questions. I can see that the problem addressed by the authors is complicated, particularly in terms of understanding the influence of subtle structural differences on material properties. Nevertheless, taking into account the originality of the undertaken problem I suggests accepting the manuscript for publication in Magnetochemistry in the present form.